# Waste Management 4.0: An Application of a Machine Learning Model to Identify and Measure Household Waste Contamination—A Case Study in Australia

**Atiq Zaman** 

Curtin University Sustainability Policy Institute, School of Design and the Built Environment, Curtin University, Bentley, WA 6102, Australia; atiq.zaman@curtin.edu.au; Tel.: +61-8-9266-9018

**Abstract:** Waste management directly and indirectly contributes to all sustainable development goals. Hence, the modernisation of the current ineffective management system through Industry 4.0-compatible technologies is urgently needed. Inspired by the fourth industrial revaluation, this study explores the potential application of waste management 4.0 in a local government area in Perth, Western Australia. The study considers a systematic literature review as part of an exploratory investigation of the current applications and practices of Industry 4.0 in the waste industry. Moreover, the study develops and tests a machine learning model to identify and measure household waste contamination as a waste management 4.0 case study application. The study reveals that waste management 4.0 offers various opportunities and sustainability benefits in reducing costs, improving efficiency in the supply chain and material flow, and reducing as well as eliminating waste by achieving holistic circular economy goals. The significant barriers and challenges involve initial investments in developing and maintaining waste management 4.0 technology, platform and data acquisition. The proof-of-concept case study on the machine learning model detects selected waste with considerable precision (over 70% for selected items). The number and quality of the labelled data significantly influences the model's accuracy. The data on waste contamination are essential for local governments to explore household waste recycling practices besides developing effective waste education and communication methods. The study concludes that waste management 4.0 can be an effective tool for acquiring real-time data; however, overcoming the current limitations needs to be addressed before applying waste management 4.0 into practice.

**Keywords:** industry 4.0; waste management 4.0; machine learning model; efficiency; waste contamination; digital waste audit; proof-of-concept

## 1. Introduction

Sustainable waste management is a global challenge and one of the most important agendas for sustainable development [1,2]. Waste management directly contributes to 12 out of the 17 sustainable development goals (SDGs), and contributes indirectly to all SDGs [1]. However, despite its dire importance in global sustainability, the relevant authorities and government bodies are not prioritising waste management at the same level as the energy, water or transport services.

Even in various developed countries, such as Australia, the resources allocated to the waste sector are insignificant. For example, the Commonwealth Government allocated only AUD 250 million for recycling modernisation and the National Waste Policy Action Plan implementation fund in Australia's 2020–2021 national budget [3]. In contrast, Australia has been experiencing a severe national waste management crisis due to the China Waste Ban since 2018, followed by the Council of Australian Governments (COAG) waste export ban (waste plastic, paper, glass and tyres) in 2020 [3]. In Australia, the state and territory governments have primary responsibility for regulating domestic waste management.

As a result, local governments predominantly provide waste management services (e.g., collection, processing and disposal) to the local communities [4].

Considering waste management as a severe problem, local governments often face various challenges, from collection, recycling and resource recovery, to the final disposal. Household waste contamination is one of these challenges for local governments and the waste recycling industry, as it causes profit losses as well as opportunities to recover resources from waste streams. Although a 6–10% waste contamination rate is considered as a standard in recycling in Australia, around 15% has been reported as the national average [5,6]. This may seem not much in terms of wasted resources, but it is significantly higher than the current import threshold set by China, which is less than 0.5%.

A recent study on reducing the contamination of household recycling found that reducing contamination necessitates infrastructure provisions and policy interventions, such as persuasion, incentives, coercion, restriction and education [7]. Although the effectiveness and cost of each intervention vary, they all primarily rely on accurate and reliable data related to current practices. For example, a proper education programme about correct recycling practices seems to be a practical and cost-effective approach to addressing household waste contamination and achieving zero waste goals [7–9]. However, the education programme, which was developed with unreliable, limited data, is often ineffective in motivating community behaviour and recycling practices.

The local government often uses waste auditing, also known as a bin-tagging programme in Australia, to collect waste data (e.g., waste composition, contamination, etc.). The bin-tagging programme is labour-intensive and covers only a limited number of households (5–10%) within the city council, owing to cost and time constraints. The data collected from the manual and time-consuming waste audit system may not represent the entire city council if the samples are not determined correctly. Moreover, there is a risk of data being obsolete if the collection and analysis take longer than usual. Therefore, the recent developments in Information and Communications Technology (ICT), Internet of things (IoT), cloud computing and smart devices pave the way to secure, reliable and real-time data, and represent catalysts for smart revolutions such as Industry 4.0.

The fourth industrial revolution, also known as Industry 4.0 (I4.0), links physical infrastructure with digital infrastructure to create a "cyber-physical system" [10]. The primary purpose of the digital transformation, and of connecting the physical world with the digital world through innovative and smart technologies (such as sensors, robotics, machine learning, etc.), is to introduce transparency into the supply system beyond the end-of-life phase using accurate and reliable data [10,11]. As a pioneer of the I4.0 concept, the German government began the computerization of the manufacturing industry in the early 2010s. Now, the application is widespread [12,13]. I4.0 has become one of the key approaches for "urban computing" and smart cities via the applying of smart gadgets, sensors and technologies to access urban big data, and for making urban service and policy decisions more effective.

As part of the development and innovation of I4.0, the machine learning model (MLM) has been discussed widely, thanks to its multifaceted benefits [14]. According to Medeiros et al. [14], amongst the various MLMs, the random forest model seems to be dominant in forecasting and predicting. There has been an influx of studies in recent years on IoT, AI, MLM, deep neural networks (DNNs) and so on, which showcase how these smart innovations contribute to fault diagnosis in power transformers, or can protect from cyber-attacks against Automated Guided Vehicles [15,16].

On the contrary, the application of AI and MLM in the waste sector is very limited and still emerging [17]. Waste management as a complex system involves numerous technical, climatic, environmental, demographic, socioeconomic and legislative parameters, and so advanced methods such as AI and MLM are required in order to tackle such complex nonlinear processes [17]. Moreover, to achieve high efficiency in eliminating waste from the system, mass personalisation by integrating hard infrastructure, data and software

platforms is also necessary to consider [18]. The uptake of mass personalisation as a service using smart technologies has already been emerging in various smart cities' initiatives.

Owing to the global interests in the smart cities agenda, the application of I4.0 in the waste industry has increased recently, particularly in the areas of waste classification [19], reduced waste generation [20], improved waste collection [21,22], supply chain [23,24], automation [25] and resource recovery [26], in order to ensure the development of a circular economy [27–29].

The notion of Waste Management 4.0 (WM4.0) has emerged from the concept of I4.0 as specifically applied to waste management [10,23]. WM4.0, in this study, refers to a "cyber–physical integrated waste management system", which connects the physical waste infrastructure (such as waste bins, collection trucks, sorting, recycling disposal infrastructure, etc.) with a digital platform through smart technologies (such as sensors, RFID, robotics, machine learning, etc.) in order to provide accurate, reliable and real-time data. WM4.0 is extremely capable in addressing the complex challenges (e.g., waste contamination) in our society, because it can provide vital data to make effective action plans and decisions enabling us to overcome current challenges.

This study aims to understand how smart and innovative solutions based on the "WM4.0" approach would provide reliable scientific data, which can be invaluable in developing an effective education programme and making correct policy decisions. The key objectives of the study are (i) to explore the current application of the WM4.0 and (ii) to develop and test a machine learning model as a WM4.0 proof-of-concept case study. The scope of the study is limited to the development of the MLM as part of the WM4.0 proof-of-concept case study. The study does not undertake a field validation through a waste audit system, and thus the volumetric reduction in waste contamination and the increase in recycling are not validated in the study.

This study identifies various benefits of applying WM4.0 in local government areas. One of its key benefits is in detecting undesirable items (contamination) in recycling bins, which will allow both households and local councils to track household waste recycling practices and be aware of any wrongdoing. Moreover, the available data would help councils identify any specific cluster and waste contamination patterns in different socioeconomic and geographical locations within the service areas, so that the council can customise its waste education and communication programme by targeting focus groups. This would save a significant amount of taxpayer money, because currently, a broad and generic waste education and communication system is offered to local residences. Furthermore, the data can be used for the benchmarking of residents' and communities' performance and gamification to improve their recycling practices. The study is important as the data would help both local councils and households improve their recycling practices, and divert more resources away from landfills by reducing the level of contamination. The economic, environment and carbon benefits are also paramount, as the recovered materials will be recirculated within the supply chain (creating new products out of recovered materials) and can help reduce the need for greenhouses, water and energy, by substituting for virgin materials.

## 2. Materials and Methods

We applied a systematic literature review (SLR) as part of an exploratory study to comprehend the current application of I4.0/WM4.0, and a WM4.0 case study of I4.0 in a local government organisation in Perth, Western Australia (WA). The following sections explicate the key aspects that were considered in the SLR and case study.

### 2.1. Systematic Literature Review (SLR)

The Scopus database is the most relevant database to this topic, and is one of the largest databases of peer-reviewed literature. Thus, the study used the Scopus Database as a primary source to conduct the systematic literature review (SLR). Since I4.0/WM4.0 is an

emerging topic, the duration of the publication was set as 2012 to 2021 (last ten years) as one of the search criteria. Articles published in English were considered during the search.

The PRISMA (Preferred Reporting Items for Systematic Reviews and Meta-Analyses) approach described by Moher et al. [30] and used to identify relevant articles was considered for this study. The PRISMA approach primarily applies four criteria: the identification of articles, screening, eligibility checks, and the assessment of the findings of the included publications. Specific search strings were used to identify the relevant publications as part of the literature review. Various terminologies, such as Industry 4.0, waste management 4.0, smart waste management automation, etc., were used. Three-level (primary, secondary and tertiary) string criteria were used to identify relevant generic (primary search string) and specific (tertiary search string) articles. The term "case study" was used in the tertiary search so that the application of I4.0/WM4.0 could be analysed based on its current applications in different case study scenarios. In each search string iteration, the number of articles on the Scopus database was recorded. After ten iterations of the search, 14,554 articles were identified (as shown in Appendix A) via the primary search criteria, followed by 633 and 103 articles via the secondary and tertiary search criteria, respectively. The list of criteria and article numbers are presented in Appendix A Table A1.

As part of the screening and eligibility check, duplicate articles (45) were excluded, and the availability of the shortlisted articles was checked against the availability of the full articles and the accessibility of the specific articles. In total, 36 articles were excluded as they were not available in full. Finally, after reviewing the full articles, 3 were excluded, as the primary focus of this review study was the study's application, so reviews of studies/literature were excluded as part of the eligibility check. Finally, 19 articles were identified for the meta-analysis of current practices. Figure 1 shows the article selection steps.

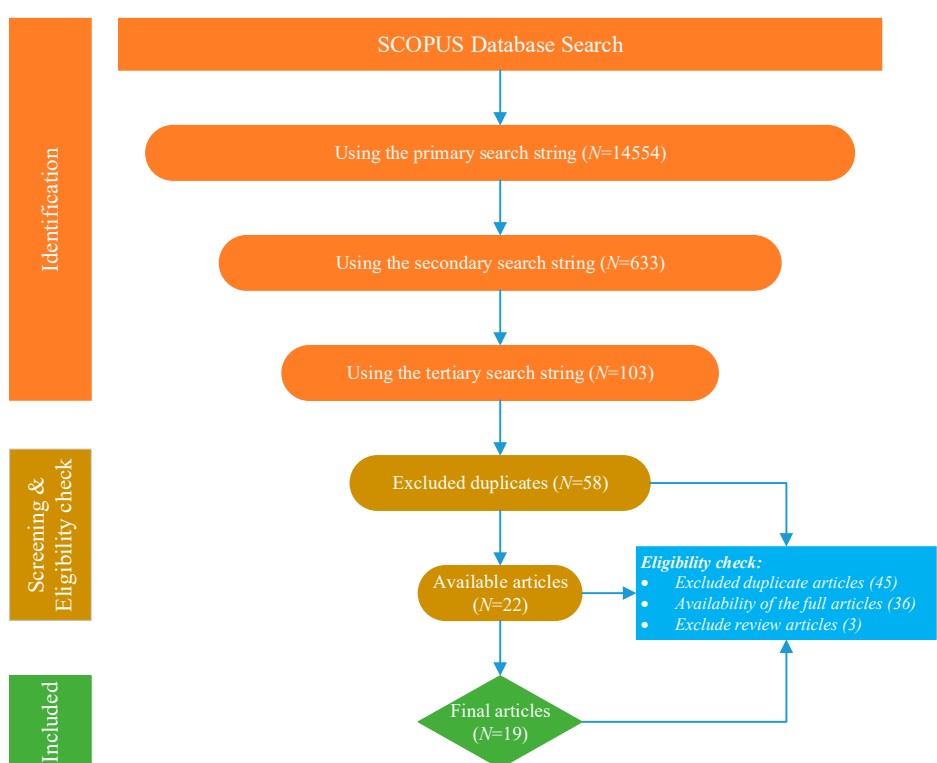

**Figure 1.** PRISMA diagram showing the selection of articles (adopted from Moher et al. [30]).

Although the duration of publication was restricted to 2012–2021, the number of publications has increased over time. Regarding the publishers of the 19 included articles, 4 were published by *Sustainability* (MDPI), followed by 3 articles published by *Computers in Industry* and 2 published by *Journal of Cleaner Production*; 10 different journals published the

remaining 10 articles. A meta-analysis reviewing the selected articles was used to identify current WM4.0 applications.

### 2.2. Waste Management 4.0 (WM4.0) Case Study

In this study, the City of Canning (a local government area in Western Australia—WA) was used as a WM4.0 case study. Although most of the city councils in WA face challenges around waste contamination, the City of Canning has taken the initiative to find ways of applying smart technologies and machine learning techniques to address the issue. Currently, truck-mounted video cameras record waste being unloaded into the trucks' hoppers and into the belly to manually check for any hazardous items, such as aerosol and batteries, so that the driver can take precautionary measures to avoid any potential fires or hazard events.

As part of the WM4.0 application, the study developed and applied a machine learning (ML) model to identify and measure household waste contamination. This study used video footage and the machine learning model to identify any particular items of interest during kerbside waste collection. The WM4.0 approach was applied in the case study by integrating a physical waste collection system through the ML model to collect digital data on waste contamination at the household level, in order to develop more effective and targeted customised future education programmes for the City of Canning based on reliable, scientific and real-time data derived from the ML model.

Figure 2 shows the conceptual application of WM4.0 in the City of Canning. Although the council provides multiple bins to residents for kerbside collection, for this particular project, the focus was placed on recycling waste, and thus we only consider the bin used for recycled waste. The MLM was used to create a database via the object detection technique, and the database can be used to identify waste contamination patterns in real time. For example, Figure 2 shows plastic bags, nappies and car batteries as examples of waste contamination that the MLM could identify.

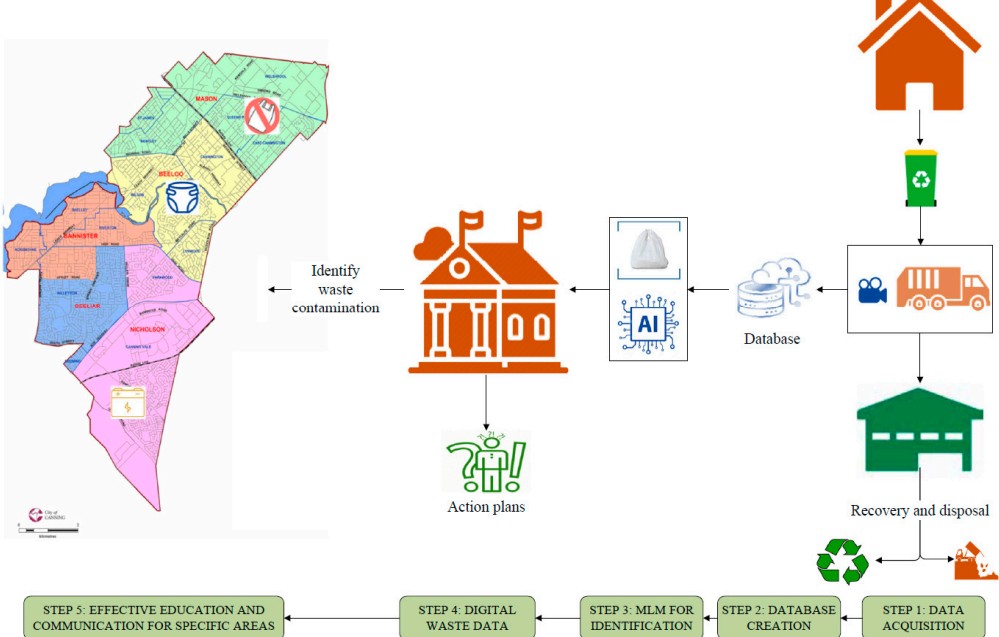

**Figure 2.** Schematic of the City of Canning Waste Management 4.0 approach.

The conceptual application of WM4.0 offers multiple benefits to the local authority, such as:

I.      Identification (frequency and severity) of contaminated waste (unacceptable items in the recycled bin (e.g., soft plastics));

II.     Identification of hazardous waste (e.g., batteries, aerosol cans, etc.);

III. Determination of patterns of incorrect recycling practices (e.g., recyclable items in a soft plastic bag) in different urban areas;

IV. Availability of real-time data (weekly, monthly and annual data) for the entire jurisdiction (currently available for selected households covered by the bin-tagging programme).

All these reliable data from the MLM would provide the city council with a great deal of understanding of the waste recycling practices in the locality. Based on scientific and reliable data, the city council can customise the waste education programme for the specific target group in specific areas, helping them to reduce waste contamination as well as to improve recycling practices.

### 2.2.1. Machine Learning Model (MLM)

The machine learning model (MLM) is a widely applied technique, particularly for image classification and detection. There is a growing interest in applying MLM in the waste industry for the prediction of waste generation [31], the identification and classification of waste [32,33], sorting [34], recycling [35] and predicting waste treatment, and recycling process [36].

Different methods are available to identify waste; the main difference between an image classifier and an object detector that is applied in the MLM is given below:

- An image classifier attempts to classify an entire image or a lone object in an image as a particular "class";
- An object detector attempts to find objects within an image and classify what each of those potentially numerous objects is.

In this case study, YOLOv4 is used as a real-time object detection framework. Many open-sourced object detection algorithms exist with different requirements and levels of accuracy. These include the YOLO, SwinTransformer, EfficientDet and R-CNN families of frameworks, among many others [37–40]. The object detection frameworks generally have a "confidence score" for multiple regions within an image.

This confidence score is essentially how confident the algorithm is that an object exists within that region. A user can set this confidence threshold and the object detector will detect any region above this threshold. Generally, object detection frameworks are designed for both speed and accuracy in order to detect objects in real time. Many of these frameworks thus make trade-offs between detection speed and accuracy. It is usually trivial to develop an object detector with slightly better accuracy that works slightly slower. A common way of comparing the variants of these object detection models is to compare their performances on the exceptionally well-labelled and -maintained MS-COCO [41] database, as a function of average precision (AP) vs. latency for a batch of 1 (time to produce a result from an input; effectively the inverse of the frame rate on real-time footage). As there are many differences between the object detection models, many hardware configurations available to run the tests on, and many ways to change the tests themselves, the metrics of any given object detection algorithm should be treated as a rough guide only. To achieve a better and more accurate performance using the MLM object detection model, various adjustments of the parameters have been considered. For example, this study applies MLM in two different trials to examine the influence on the quantity and quality of the labelled data. In the first trial (Trial 1), only one specific item in each image was labelled, whereas in Trial 2 all selected items (six items) were labelled based on their presence in the image. The following key steps were considered in the WM4.0 case study. Moreover, some of the objective items have been combined into broader categories, as shown below, in order to comprehend how well the MLM would predict following this parameter change. This parameter change also helps with the issue of overfitting in the predicted model. Although the quantity of training data affects overfitting, in our study, it is apparent that the quantity of quality data matters more than the quantity of training data, as discussed in Section 4.1. This study applies various "trial and error" approaches to derive the most desirable outcomes from the model.

### 2.2.2. Development of a Database of Labelled Images

A total of 20,491 of the 25,791 labelled images were annotated, in six different waste categories. A breakdown of the images is given below in Table 1. Several labels were reclassified to increase the amount of data per class and the viability of the YOLOv4 model:

- "PET Bottle" labels were combined with "Plastic Container" labels;
- "Pizza Box" labels were combined with "Paper/Cardboard" labels;
- "Soft Plastic Packaging" labels were discarded, as sufficient data were not available.

**Table 1.** Breakdown of labels used in initial data.

| Object Label | Number of Labelled Images in Trial 1 | Number of Labelled Images in Trial 2 |
|---|---|---|
| Plastic Container | 10,006 | 583 |
| Soft Plastic Bag | 5035 | 1294 |
| Pizza Box | 3140 | 83 |
| Tetra-Pack | 3049 | 264 |
| PET Bottle | 2535 | 248 |
| Paper/Cardboard | 2026 | 1907 |

Table 1 shows the number of labelled images used in the MLM for Trials 1 and 2. Generally, only 1 or 2 items are labelled in any given image, up to the maximum of 7 items labelled in one image. In Trial 1, priority was given to the number of labelled items as only the minimum number of items (only one of two) was labelled in the images. Not labelling every data point can reduce the performance of YOLOv4 models.

In the second trial, every single item was labelled in each image based on its presence, in order to improve the quality of the labelled images. In total, 1549 unique images were received with 4379 labels, broken down by class in Table 2. While far fewer data are available, their quality is far higher. We will see later that the models achieve improved performance even when only running on this smaller amount of data.

**Table 2.** The key concepts and analytical approaches considered in the literature.

| Articles | Key I4.0/WM4.0 Concepts Covered in the Literature | | | | | | | | Analytical Approach | | |
|---|---|---|---|---|---|---|---|---|---|---|---|
| | Circular Economy | Supply Chain | Lean Manufacturing | Material/Waste Flow | Smart/Cloud System | Reverse Logistics | Smart/Z WM | Sustainability—Including SDGs, TBL | Descriptive/Conceptual | Simulation | Experimental/Implementation |
| Amjad et al., 2021 [42] | | | ✓ | | | | | ✓ | ✓ | ✓ | |
| Belaud et al., 2019 [43] | | | | | | | | ✓ | ✓ | | |
| Cotet et al., 2020 [44] | | | | | ✓ | | | | | | ✓ |
| de Sousa Jabbour et al., 2018 [45] | ✓ | | | ✓ | ✓ | | | ✓ | ✓ | | |
| Fatimah et al., 2020 [46] | ✓ | | | | | | ✓ | ✓ | ✓ | | |
| Fisher et al., 2020 [47] | ✓ | | | | | | ✓ | | | | ✓ |
| Garrido-Hidalgo et al., 2019 [48] | ✓ | | | | | ✓ | | | | | ✓ |
| Goodall et al., 2019 [49] | ✓ | | | | ✓ | | | | | ✓ | |
| Gu et al., 2017 [50] | | | | | ✓ | | | | ✓ | | |

**Table 2.** *Cont.*

| Articles | Key I4.0/WM4.0 Concepts Covered in the Literature | | | | | | | | Analytical Approach | | |
|---|---|---|---|---|---|---|---|---|---|---|---|
| | Circular Economy | Supply Chain | Lean Manufacturing | Material/Waste Flow | Smart/Cloud System | Reverse Logistics | Smart/Z WM | Sustainability—Including SDGs, TBL | Descriptive/Conceptual | Simulation | Experimental/Implementation |
| Ilari et al., 2021 [51] | | | | | √ | | | √ | √ | | |
| Kerdlap et al., 2019 [52] | | | | | | | √ | √ | √ | | |
| Krishnan et al., 2021 [53] | | | | | √ | | √ | | √ | | |
| Massaro et al., 2021 [54] | √ | | | | | | | | √ | | |
| Mastos et al., 2020 [55] | | √ | | | | | | | | | √ |
| Oltra-Mestre et al., 2021 [56] | | √ | | | | | | | √ | | |
| Schoeman et al., 2021 [57] | | | √ | √ | | | √ | | √ | | |
| Tran et al., 2021 [58] | | | √ | | | | | | | | √ |
| Wang and Wang, 2017 [59] | | | | | √ | | | | | | √ |
| Xing et al., 2020 [60] | √ | | | √ | | | √ | | | | √ |
| Total coverage | 7 | 2 | 3 | 2 | 7 | 2 | 6 | 6 | 11 | 2 | 7 |

2.2.3. Development of the ML Algorithm/Full YOLOv4 Object Detection Pipeline

Several data pipeline models were used for the analysis. First, a baseline model was only used on the original and empty truck data. Then, a model on the original data and the empty truck data with primary image augmentation was applied (as shown in Figure 3). Finally, a model using data from steps 1 and 2 and empty truck data, all with basic image augmentation, was analysed. The initial pre-trained weights were the output of the training of the same YOLOv4 object detection algorithm used on Data Pipeline 1. The weights that performed best for this were obtained using early stopping, likely due to the much more extensive data than was available in Pipeline 1. Methods 1 and 3 worked the best out of all data pipeline models, and are the only ones reported in the result section.

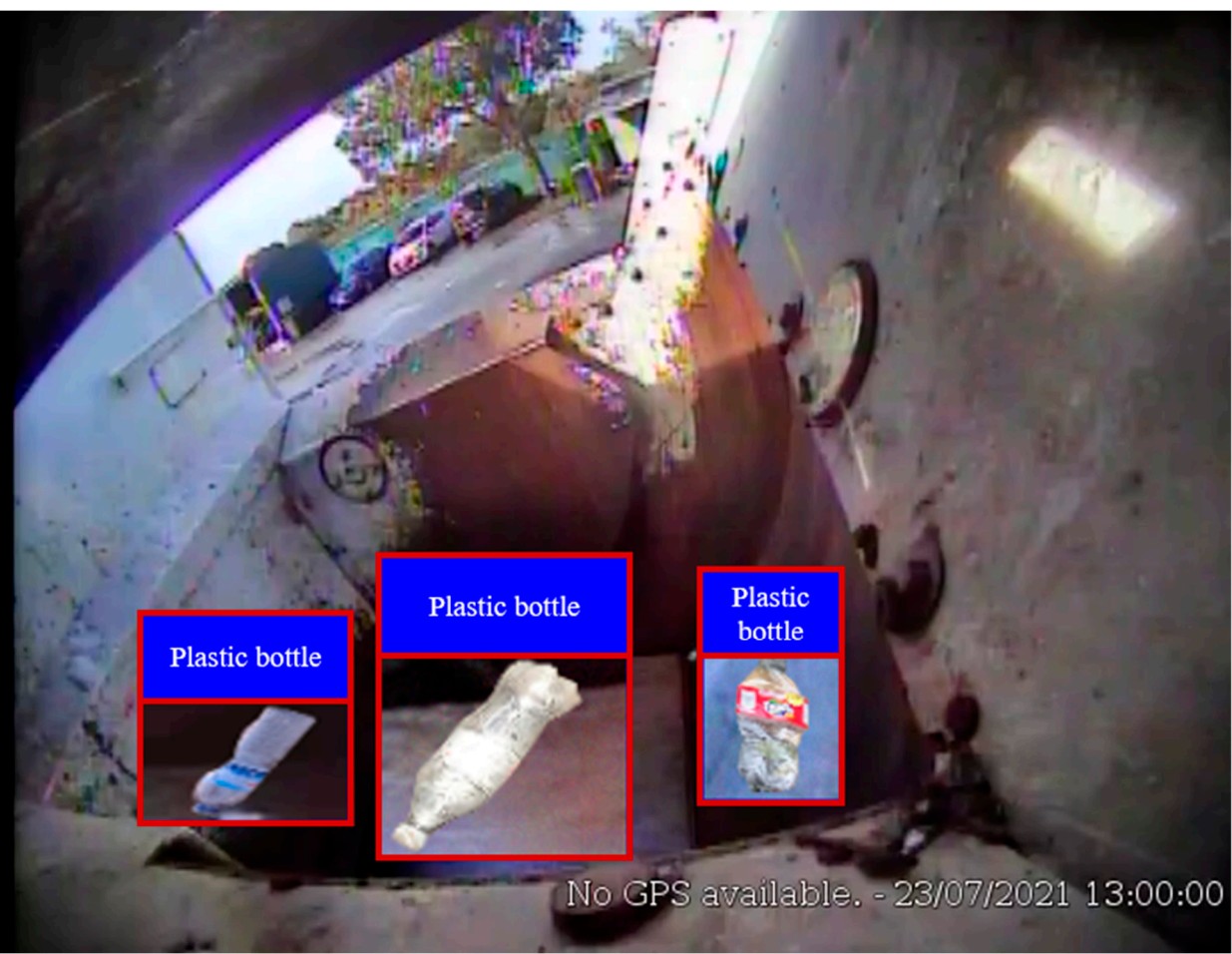

**Figure 3.** Extra data inserted into the recycling truck.

## 3. Current Application of I4.0/WM4.0

The qualitative meta-analysis of the 19 shortlisted articles (presented in Table 2) revealed the application of I4.0 or WM4.0 in different industries, including agriculture, food-processing, electronic waste, municipal solid waste, heavy industry such as iron and steel, automobile, scrap metal, manufacturing, wastewater, etc. The key findings from the literature review are presented, outlining the core concepts, analytical approaches and technologies used, and reporting the opportunities and limitations related to applying WM4.0.

### 3.1. Key Concepts and Analytical Approaches

Although the key focus when assessing the articles was on waste management, diverse concepts were identified in the literature. Circular economy (seven articles), smart/cloud system (seven articles), smart waste management (six articles) and sustainability aspects (six articles) were the most applied concepts in relation to I4.0/WM4.0, followed by lean manufacturing (three articles), supply chain (two articles), material flow (two articles) and reverse logistics (two articles). Regarding the analytical approach, most of the literature (11 out of 19 studies) employed descriptive or conceptual frameworks, 2 studies used simulation model and 7 studies used experimental or proof-of-concept experiments.

### 3.2. Applied I4.0/WM4.0 Technologies

Some form of an ICT perception framework was the most widely used technique (15 out of 19 studies), followed by IoT networks. The ICT application was used in five studies,

and the same number of studies applied RFID, BLE and WSN or similar technologies, as shown in Table 3.

**Table 3.** The most common I4.0/WM4.0 technologies used.

| Articles | Key I4.0/WM4.0 Technologies Used in the Literature | | | |
|---|---|---|---|---|
| | ICT Perception Framework | ICT Application | IoT Network—Sensors | IoT Network—BLE/RFID/WSN |
| Amjad et al., 2021 [42] | √ | √ | √ | |
| Belaud et al., 2019 [43] | √ | | | |
| Cotet et al., 2020 [44] | | | √ | |
| de Sousa Jabbour et al., 2018 [45] | √ | | | |
| Fatimah et al., 2020 [46] | √ | | √ | |
| Fisher et al., 2020 [47] | | √ | | √ |
| Garrido-Hidalgo et al., 2019 [48] | √ | | √ | √ |
| Goodall et al., 2019 [49] | | √ | √ | |
| Gu et al., 2017 [50] | √ | | √ | √ |
| Ilari et al., 2021 [51] | √ | | | |
| Kerdlap et al., 2019 [52] | √ | | √ | |
| Krishnan et al., 2021 [53] | √ | | | |
| Massaro et al., 2021 [54] | √ | | | |
| Mastos et al., 2020 [55] | | √ | √ | √ |
| Oltra-Mestre et al., 2021 [56] | √ | | | |
| Schoeman et al., 2021 [57] | √ | | | |
| Tran et al., 2021 [58] | √ | | √ | |
| Wang and Wang, 2017 [59] | √ | | | |
| Xing et al., 2020 [60] | √ | √ | √ | √ |
| Total coverage | 15 | 5 | 10 | 5 |

### 3.3. Reported Opportunities and Limitations

The selected studies reported various benefits and opportunities related to applying I4.0/WM4.0. The most common opportunities included improvements in the overall efficiency [50], reductions in cost, enhancements in the profit [42,46,52,57], efficiency and optimization in the supply chain [55], resource recovery [46,48], the elimination of waste [53,57,58] and data visibility [56].

Considerations of social aspects were rarely reported, as the key focus in most of the studies was on the technologies, rather than on translating how these I4.0/WM4.0 technologies would impact the greater societal system. Fatimah et al. [46] considered the consumer perceptions and wellbeing of scavengers, and Kerdlap et al. [52] considered the use of social networks as a way of benefiting multi-stakeholders while decoupling economic growth.

Sustainability benefits were reported in several studies. For example, Amjad et al. [42], Belaud et al. [43], Fatimah et al. [46], Garrido-Hidalgo et al. [48], Ilari et al. [51] and Xing et al. [60] reported emission and pollution reductions, sustainability assessments, and the conservation of resources as the key environmental benefits of using I4.0/WM4.0.

Concerning the limitations and the key constraints related to the application of I4.0/WM4.0, the data acquisition and initial investments for developing and maintaining the I4.0/WM4.0 application were reported as the most significant constraints [44,47,49–51]. In addition, several studies emphasised that the negative impacts of I4.0/WM4.0 on society (e.g., automation, joblessness, etc.) need more attention and further investigation [42,43]. Furthermore, trust, privacy, and protecting commercial interests were also reported [50,55,58] as constraints due to the virtual presence of sensitive information. Moreover, as an emerging field, various technological limitations were also reported in the context of applying I4.0/WM4.0, and these primarily include the lack of a generalised concept, the need for industry-specific solutions, and the lack of physical resources [19,53,55].

## 4. WM4.0 Case Study

### 4.1. MLM Model

In a proof-of-concept study, the MLM successfully identified waste items from the video footage with a considerable level of accuracy. However, the accuracy level varied based on which type of waste was identified. Figure 4 shows the waste item identified during the testing of MLM. The following section explains and discusses the key findings of the MLM.

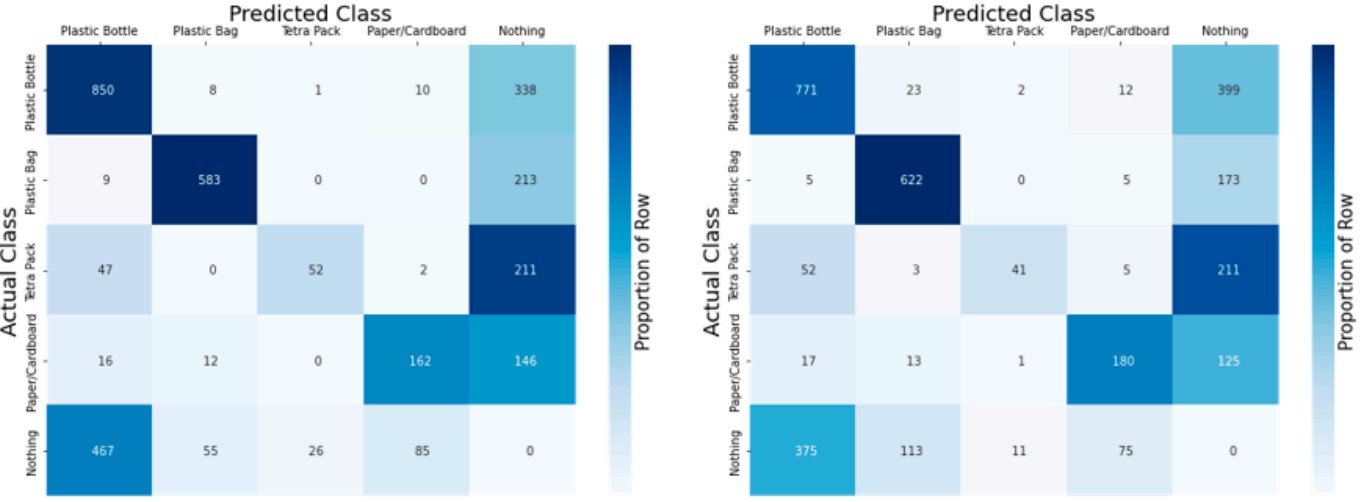

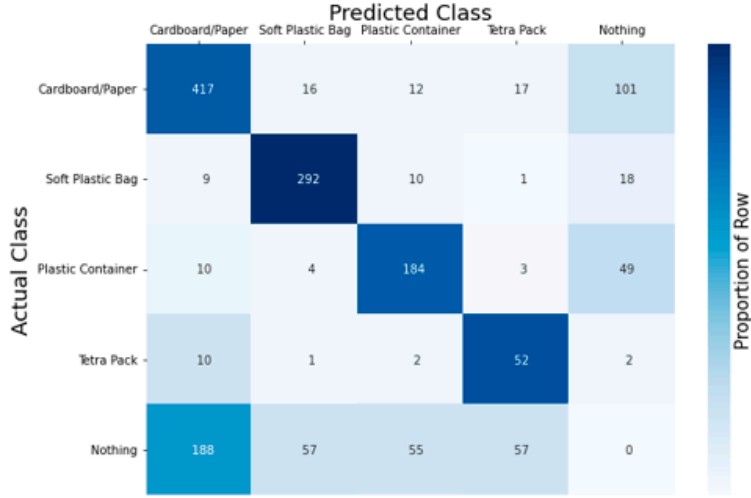

**Figure 4.** Data Pipeline model 1 (**a**) and 2 (**b**) in Trial 1 and Data Pipeline model 1 in Trial 2 (**c**).

As outlined in the methodology, two pipeline models were tested in case study Trial 1. Overall, the models were quite similar, indicating a non-dependence on the training method. As seen below (Figure 4), the F1 scores of the normalised averaged class precisions and recalls were the same (0.65). However, there were some differences in the specific class APs, indicating a preference for one model over another. Confusion matrices for both Data Pipeline 1 and Data Pipeline 2 are presented to show the difference between the two Data Pipelines. The confusion matrices below were calculated using the models trained on the initial data that was evaluated in relation to the new labelled data.

With a confidence threshold of 0.25 for identifying an object, the metrics for the Data Pipeline 1 model and Data Pipeline 3 models were calculated using a custom identifier based on whether a prediction overlapped with an actual class with an IoU of at least 0.25 (Table 4).

**Table 4.** The average precision data derived from the MLM-Trial 1.

| Trials | Data Model | Average Class Precision | Average Class Recall | F1-Score | MAP @ 0.5 IoU Threshold |
|---|---|---|---|---|---|
| Trial 1 | Pipeline 1 model | 0.62 | 0.87 | 0.72 | 0.50 |
| | Pipeline 3 model | 0.61 | 0.85 | 0.71 | 0.54 |
| | Data Model | Plastic Container AP | Plastic Bag AP | Tetra-Pack AP | Paper/Cardboard AP |
| | Pipeline 1 model | 60.77% | 72.93% | 24.82% | 42.97% |
| | Pipeline 3 model | 57.74% | 76.62% | 23.39% | 47.82% |
| Trial 2 | Data Model | Average Class Precision | Average Class Recall | F1-Score | MAP @ 0.5 IoU Threshold |
| | Pipeline 1 model | 0.78 | 0.71 | 0.80 | 0.62 |
| | Data Model | Plastic Container AP | Plastic Bag AP | Tetra-Pack AP | Paper/Cardboard AP |
| | Pipeline 1 model | 65.44% | 82.56% | 37.14% | 62.87% |

The table shows that the precision of the detection of images varied by a considerable level. The average precision of plastic containers was within 57–61%, that of plastic bags was within 73–77%, tetra packs within 23–25%, and the detection of paper/cardboard was within 43–48%. As a proof-of-concept study, the results are promising.

In Trial 2, a new YOLOv4 model was trained on a training subset of 1549 new images (roughly 1200 annotated images) using only Data Pipeline 1. The image identification improved in Trial 2 despite the quantity of labelled data being lower than that of labelled data used in Trial 1.

In comparing the ratios, the first models had F1 scores of 0.72 and 0.71 (in Table 4), while the model trained on the new data had an F1 score of 0.80 (Table 4), showing a clear improvement despite the much lower quantity of training data used. Although the use of at least 2000 labelled images is recommended for YOLOv4, Trial 2 of the case study shows that the quality of the labelled images matters, and better-quality data determine a higher precision level than the higher quantity of lower-quality labelled data used in Trial 1.

The most important consideration when using YOLOv4 training data is that all objects of interest must be identified in every image. YOLOv4 will still work as an object detector, but its accuracy may be reduced. Figure 4b is an example of how there are multiple types of relevant objects in an image (cardboard, tetra packs, etc.), but only a single plastic container is labelled. YOLOv4 is trained not only on what is labelled, but also implicitly on what is not labelled (i.e., it trains itself on what it should not detect as well as what it should detect). Having objects of interest not labelled can therefore lower the accuracy of any trained YOLOv4 model. Some of the observations made while developing and testing the models using object detector YOLOv4 are given below:

- At least 2000 images arise per class of object one wants to detect;

- All objects of interest are identified in every image;
- Examples of objects not of interest are present in some images but are not labelled;
- At least 2000 examples of context-appropriate images arise with no items of interest (no labels);
- Objects arise in different orientations, positions, contexts, etc. At least one object arises that is similar to objects in the training dataset, with roughly the same shape, side shown, relative size, angle of rotation, illumination, background, etc.;
- Sufficient image resolution/information must be supplied for each object in order to differentiate it from the background and other objects.

### 4.2. The Key Considerations for Applying MLM

It is evident from the literature review that I4.0 gives rise to a wide range of techniques and capabilities related to tracking, assessing and monitoring complex urban services, such as waste management. However, consideration should be given to how innovative tools such as MLM can be applied appropriately in addressing urban problems. As a proof-of-concept study, this MLM shows potential for use in the identification of waste contamination using the objective detection technique. It is also evident from this proof-of-concept study that MLM is not fool-proof, as it has certain limitations. Thus, the model needs to overcome certain limitations related to labelled and trained data in order to accurately identify the objects. The following factors need to be considered in overcoming the current limitations of the model:

I.     More appropriate data input—Having every object of interest labelled in every input image may improve the results. Including extra data and sources for objects that can appear in trucks but are not often seen in recycling as subsets of other items, such as international waste in the TACO dataset, would increase the reliability, but not necessarily the accuracy;

II.     Higher resolution imagery—Thanks to the 640 × 480 pixel resolution zoomed-out view, many of the actual objects are very pixelated, and objects of interest can approach 10–25 pixels across, which makes them hard to resolve with a single frame, even for a human. A few solutions are possible, such as using a higher-resolution camera and optically zooming the camera into the hopper;

III.     Images under different conditions—YOLOv4 can only learn from what the model has seen, and generalisation is an incredibly hard problem to tackle with modern ML methods. Images from different camera angles, and using different lighting conditions and objects of different sizes and orientations, would help improve the accuracy;

IV.     Model choices—For the purposes of this study, a simple YOLOv4 model was used. However, there is the possibility of using different models to boost the accuracy.

### 4.3. Opportunities and Constraints in Applying WM4.0 in the Local Government

Although the current model displays some limitations, there are significant opportunities available for further developing the MLM for WM4.0 applications at the local council level, such as in the City of Canning. The biggest constraint is the initial investment required for the city to develop the model, as well as the IoT platform. However, considering the current practices for identifying waste contamination used by the council, such as using an intermittent physical waste audit system that involves a very limited number of households (mostly 5–10%), and the associated cost of conducting such a bin-tagging programme, the further development of the model seems more logical, as it would save more money in the long-run. The current waste education programme run by the council is based on a very limited quantity of sample data, with limited considerations of residents' varied socioeconomic and cultural backgrounds.

The improved MLM would provide accurate and reliable data for every household, but it would also provide data in real-time. Therefore, local governments can customise their waste education programme based on accurate data, and communicate with the

respective residents accordingly. Moreover, WM4.0 would enable the creation of a better platform for communicating to and motivating residents regarding their recycling behaviour and practices.

With the current "bin tagging" programme, the local council can only achieve a very small sample size (5–10%) of total households, and only at a certain time (once–twice a year). However, under the MW4.0 concept, local governments can acquire data 24/7 for the entire jurisdiction. Therefore, the possibility of improving the current waste education approach and motivating the local community towards reducing waste contamination is massive. Although data privacy could be an issue, the risks associated with privacy concerns could be overcome by applying proper strategies and maintaining guidelines.

## 5. Conclusions

The level of waste contamination in Australia is very high, and due to waste contamination, valuable resources are lost, consequently creating economic and environmental burdens. Unfortunately, the data related to waste contamination are very limited, and access to data is possible only though the traditional waste audit system, which is often costly and time-consuming. WM4.0 (inspired by I4.0) has potential use in accessing waste contamination data for entire service areas for the entire year. Therefore, to reduce household waste contamination levels, this study tried to explore the current application of the I4.0 concept in the waste management system through a systematic literature review. In addition, the study developed an MLM as a WM4.0 case study to consider it for potential use by local governments in Australia to measure their waste contamination level.

It is evident from the SLR that, as an emerging field, the application and consideration of I4.0/WM4.0 is still limited, but it is increasing with time. The study found that WM4.0 is used in different areas of waste management, from the supply chain, sorting and resource recovery, to the final processing. WM4.0 offers various opportunities and benefits in relation to reducing costs, improving efficiency in the supply chain and material flow, and reducing and eliminating waste from society by achieving holistic circular economy goals. The study also identified various social and environmental benefits, such as improving livelihood, being socially conscious, and reducing emissions and pollution, in relation to WM4.0. However, the initial investments required for developing and maintaining MW4.0, and for data acquisition, were identified as the major challenges in applying I4.0/WM4.0.

An MLM was developed as a proof-of-concept, and was used to identify waste objects or contamination. Although the average precision of object detection varied for different waste items, the results were promising. By overcoming the current limitations, WM4.0 can be applied to identify the level of waste contamination in the local government area. Based on the case study's findings, we conclude that the City of Canning would be benefited in the long run by embracing WM4.0 in its current waste management practices. The greatest benefits of entering the digital waste audit system using WM4.0 would be the availability of accurate real-time data, which is vital in overcoming current waste management challenges and planning for a better future.

The findings of this study can provide a better understanding of the current application of I4.0/WM4.0 in real-world scenarios. In addition, drawing from a case study of a local city council (the City of Canning) in Western Australia, this study's results would be of interest for and beneficial to the local government authorities when considering their WM4.0 strategy in the future. Moreover, this preliminary study will increase the confidence of local councils in investing in the project and developing the model further, up to a full-scale application in the future.

The scope of this study was limited to an exploration of the current application of I4.0 in waste management and the development of an MLM as a proof-of-concept case study. Therefore, the study did not consider validation through an actual waste audit system. This is a limitation of the study, as the scope was to develop and test an MLM model, rather than validating it in the field. Future studies should focus on expanding the labelled items

and validating them through actual filed audits, so that the precision of the data provided by the MLM can be validated.

The author acknowledges that this study has some limitations, such as considering only specific years (last 10 years) when identifying relevant articles for the SLR. Moreover, the model's limitations, such as the quality of the images, were beyond the scope of the current study. Future studies should consider the key recommendations outlined to overcome these limitations.

**Funding:** This research and the APC was funded by the City of Canning, grant number RES-63426.

**Informed Consent Statement:** Not applicable.

**Acknowledgments:** The research project was jointly conducted by the Curtin University Sustainability Policy Institute (CUSP) and the Curtin Institute for Computation (CIC). The author wants to thank Leigh Tyers and Daniel Marrable from CIC for technical support in the project. The author also wants to thank three anonymous reviewers for their valuable feedback related to improving the quality of the manuscript.

**Conflicts of Interest:** The authors declare no conflict of interest.

## Abbreviations

| Acronym | Full Form |
|---|---|
| AP | Average Precision |
| BLE | Bluetooth Low Energy |
| DNN | Deep Neural Network |
| I4.0 | Industry 4.0 |
| ICT | Information and Communications Technology |
| IoT | Internet of Things |
| ML | Machine Learning |
| MLM | Machine Learning Model |
| MS-COCO | Microsoft Common Objects in Context |
| PRISMA | Preferred Reporting Items for Systematic Reviews and Meta-Analyses |
| RFID | Radio-Frequency Identification |
| R-CNN | Region-Based Convolutional Neural Networks |
| SDGs | Sustainable Development Goals |
| SLR | Systematic Literature Review |
| WA | Western Australia |
| WM4.0 | Waste Management 4.0 |
| WSN | Wireless Sensor Network |
| YOLOv4 | You Only Look Once |

## Appendix A

**Table A1.** Identification of relevant publications from Scopus database using a systematic search approach.

| Search Attempt | Search String Used and Number of Articles Found in Each Attempt | | | | | |
|---|---|---|---|---|---|---|
| | Primary Search String Criteria | No. of Articles | Secondary Search String Criteria | No. of Articles | Tertiary Search String Criteria | No. of Articles |
| 1 | industry 4.0 OR waste management 4.0 | 4745 | industry 4.0 OR waste management 4.0 AND smart technology | 161 | industry 4.0 OR waste management 4.0 AND smart technology OR circular economy AND case study | 26 |
| 2 | industry 4.0 OR waste management 4.0 OR zero waste | 4875 | industry 4.0 OR waste management 4.0 OR zero waste AND waste data | 88 | industry 4.0 OR waste management 4.0 OR zero waste AND waste data AND case study | 23 |

**Table A1.** *Cont.*

| Search Attempt | Search String Used and Number of Articles Found in Each Attempt | | | | | |
|---|---|---|---|---|---|---|
| | Primary Search String Criteria | No. of Articles | Secondary Search String Criteria | No. of Articles | Tertiary Search String Criteria | No. of Articles |
| 3 | industry 4.0 OR waste management 4.0 OR circular economy | 1262 | industry 4.0 OR waste management 4.0 OR circular economy AND smart technology | 79 | industry 4.0 OR waste management 4.0 OR circular economy AND smart technology AND case study | 12 |
| 4 | smart waste management | 1822 | smart waste management AND automation | 132 | smart waste management AND automation AND case study | 11 |
| 5 | automation in waste management | 621 | automation in waste management AND smart technology | 71 | automation in waste management AND smart technology AND case study | 7 |
| 6 | industry 4.0 OR waste management 4.0 OR automation in waste management | 319 | industry 4.0 OR waste management 4.0 OR automation in waste management AND big data | 23 | industry 4.0 OR waste management 4.0 OR automation in waste management AND big data AND case study | 9 |
| 7 | waste management 4.0 AND Internet of thing | 72 | waste management 4.0 AND Internet of thing AND smart technology | 15 | waste management 4.0 AND Internet of thing AND smart technology AND case study | 4 |
| 8 | waste management AND Internet of thing OR industry 4.0 | 45 | waste management AND Internet of thing OR industry 4.0 AND smart technology | 15 | waste management AND Internet of thing OR industry 4.0 AND smart technology AND case study | 4 |
| 9 | waste management 4.0 AND Internet of thing | 43 | waste management 4.0 AND Internet of thing AND circular economy | 6 | waste management 4.0 AND Internet of thing AND circular economy AND case study | 4 |
| 10 | waste management AND IoT OR I4.0 | 750 | waste management AND IoT OR I4.0 AND database | 43 | waste management AND IoT OR I4.0 AND database AND case study | 3 |
| Total | Primary criteria | 14,554 | Secondary criteria | 633 | Tertiary criteria | 103 |
| After excluding the duplication | | | | | | 58 |
| Reviewing the document's availability and relevance | | | | | | 22 |
| Number of identified articles for analysis after excluding the review articles | | | | | | 19 |

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
