# Peer review of "Waste Management 4.0: An Application of a Machine Learning Model to Identify and Measure Household Waste Contamination—A Case Study in Australia"

_sustainability, doi:10.3390/su14053061_

Round 1

Reviewer 1 Report

The quality of the article and the structured review of the literature is impressive. The research behind this publication involved a great deal of effort and is therefore particularly valuable.

For the reader, it is not clear until the end what the impact is. Neither at the beginning nor at the end of the article was it shown how many tonnes of waste can be separated or recycled. As a result, it remains unclear how much CO2 is ultimately not produced by waste incineration or in landfills as a result of this waste. It is strongly recommended that a quantitative analysis be carried out to substantiate the importance of the article in this regard.

Author Response

Reviewer comment:

The quality of the article and the structured review of the literature is impressive. In addition, the research behind this publication involved a great deal of effort and is therefore particularly valuable.

Author’s response:

  • The author would like to thank the reviewer for finding the literature's structure impressive and the meta-analysis's findings valuable.

Reviewer comment:

For the reader, it is not clear until the end what the impact is. Neither at the beginning nor at the end of the article was it shown how many tonnes of waste can be separated or recycled. As a result, it remains unclear how much CO2 is ultimately not produced by waste incineration or in landfills as a result of this waste. It is strongly recommended that a quantitative analysis be carried out to substantiate the importance of the article in this regard.

Author’s response:

  • Thank you for your feedback. The author has incorporated the impact statement in the Introduction as well as in conclusion to clarify the significance of the study. The study has developed and tested a machine learning model to detect waste contamination by identifying certain waste items but the model and analysis didn’t cover the exact quantity were saved; thus the environmental benefits of diverting waste from landfills and saving CO2 emissions were also not covered in the study. A more precise scope of the study is included in the revised manuscript.

English language: a professional proofread was employed for the language correction.

Reviewer 2 Report

I am glad the authors made this valuable study using a machine-learning model concerning waste management. Some suggestions are listed as follow.

Abstract: It is advised to avoid using acronyms such as I4.0 and SLR in the abstract, and therefore using full name should be sufficient. Also, it looks like waste management 4.0 (MW4.0) in the Abstract should be (WM4.0).

The authors may wish to use keywords efficiently, primarily waste in Waste Management and Waste contamination. Also, Industry 4.0 and Waste Management 4.0.

Page 1 Line 8 Please use articles (a,  an, the) when required. For instance, if you refer to the Sustainable Development Goals (SDGs), which the UN introduced, you may update the first sentence.

Page 2 Line 43: “since 2018 and 2020”, it looks like mentioning 2020 is not required.

The Introduction may cover more general studies than those limited to Australia and China. In addition, if the authors like to emphasise the proof-of-concept, the title can be modified by adding “a case study in Australia”.

It is worth mentioning scaleable personalisation as a long term strategy to eliminate waste. “Mass Personalisation as a Service in Industry 4.0: A Resilient Response Case Study.” is suggested since this study is in the context of Industry 4.0.

The presentation of Table 2 can be improved. Align left the first column. Also, you may change the text in the first row from horizontal to vertical.

Figure 4: It is recommended to add a legend to determine classes.

Table 4: distinguishing between Trial 1 and 2 is not clear. Please apply a suitable frame.

This study is a good representation of using machine learning to address a severe challenge than utilising a wide range of Industry 4.0 technologies and capabilities. So, you may wish to elaborate on this in subsection 4.2.

I hope you found the feedback helpful.

Stay safe!

Author Response

I am glad the authors made this valuable study using a machine-learning model concerning waste management. Some suggestions are listed as follow.

Author’s response:

  • The author would like to thank the reviewer for finding the study valuable

Abstract: It is advised to avoid using acronyms such as I4.0 and SLR in the abstract, and therefore using full name should be sufficient. Also, it looks like waste management 4.0 (MW4.0) in the Abstract should be (WM4.0).

  • Thank you for your advice and the acronyms are removed from the Abstract.

The authors may wish to use keywords efficiently, primarily waste in Waste Management and Waste contamination. Also, Industry 4.0 and Waste Management 4.0.

  • As suggested the keywords are updated.

Page 1 Line 8 Please use articles (a,  an, the) when required. For instance, if you refer to the Sustainable Development Goals (SDGs), which the UN introduced, you may update the first sentence.

Page 2 Line 43: “since 2018 and 2020”, it looks like mentioning 2020 is not required.

  • Thanks for the suggestion and they are corrected in the revised version.

The Introduction may cover more general studies than those limited to Australia and China. In addition, if the authors like to emphasise the proof-of-concept, the title can be modified by adding “a case study in Australia”

  • The title of the article has been modified based on the suggestion.

It is worth mentioning scaleable personalisation as a long term strategy to eliminate waste. “Mass Personalisation as a Service in Industry 4.0: A Resilient Response Case Study.” is suggested since this study is in the context of Industry 4.0.

  • The suggested article is cited in the revised manuscript by emphasising scalability and mass personalisation as a service.  

The presentation of Table 2 can be improved. Align left the first column. Also, you may change the text in the first row from horizontal to vertical.

Figure 4: It is recommended to add a legend to determine classes.

Table 4: distinguishing between Trial 1 and 2 is not clear. Please apply a suitable frame.

  • The author thanks the reviewer and the suggested changes in tables and figures have been done in the revised version

This study is a good representation of using machine learning to address a severe challenge than utilising a wide range of Industry 4.0 technologies and capabilities. So, you may wish to elaborate on this in subsection 4.2.

  • An elaborated discussion is provided in subsection 4.2.

English language: a professional proofread was employed for the language correction.

Reviewer 3 Report

The topic Waste Management 4.0: an application of a machine-learning model to identify and measure household waste contamination is potentially interesting, however, there are some issues that should be addressed by the authors: The Introduction" sections can be made much more impressive by highlighting your contributions. The contribution of the study should be explained simply and clearly. The authors should further enlarge the Introduction with current works based on artificial intelligence to improve the research background, for example: Effective IoT-based Deep Learning Platform for Online Fault Diagnosis of Power Transformers Against Cyberattack and Data Uncertainties, Effective multi-sensor data fusion for chatter detection in milling process‏; Effective feature selection with fuzzy entropy and similarity classifier for chatter vibration diagnosis; Development of an IoT Architecture Based on a Deep Neural Network against Cyber Attacks for Automated Guided Vehicles‏.

Clarify how you adjust the parameters of the proposed algorithm

Clarify how you handle the overfitting

Increase the quality of the figures

Conclusion section should be rearranged. According to the topic of the paper, the authors may propose some interesting problems as future work in the conclusion.

Author Response

Reviewer’s comment:

The topic Waste Management 4.0: an application of a machine-learning model to identify and measure household waste contamination is potentially interesting, however, there are some issues that should be addressed by the authors:

Author’s response:

  • The author wants to thank you for finding the article interesting and for providing valuable feedback. The author has revised the article by addressing the reviewer’s comments as below.

The Introduction" sections can be made much more impressive by highlighting your contributions. The contribution of the study should be explained simply and clearly. The authors should further enlarge the Introduction with current works based on artificial intelligence to improve the research background, for example: Effective IoT-based Deep Learning Platform for Online Fault Diagnosis of Power Transformers Against Cyberattack and Data Uncertainties, Effective multi-sensor data fusion for chatter detection in milling process‏; Effective feature selection with fuzzy entropy and similarity classifier for chatter vibration diagnosis; Development of an IoT Architecture Based on a Deep Neural Network against Cyber Attacks for Automated Guided Vehicles‏.

Author’s response:

  • Thank you for this valuable feedback. The contribution of the study has been incorporated in simple and straightforward statements. Moreover, the current work based on deep learning to identify and its application has been revised and strengthened in the revised manuscript.

Review comment:

Clarify how you adjust the parameters of the proposed algorithm

Author response:

  • The approach of adjusting the parameters of the proposed algorithm has been clarified in section 2.2.1.

Review comment:

Clarify how you handle the overfitting

Author response:

  • The clarification is included in section 2.2.1, which explains how to handle overfitting.

Review comment:

Increase the quality of the figures

Author response:

  • Some of the image quality is low as the video footage used in the MLM was not high resolution. This has been identified as the limitation of the study and the proposed model that a high-resolution image is needed.

Reviewer comment:

The conclusion section should be rearranged. According to the topic of the paper, the authors may propose some interesting problems as future work in the conclusion.

Author response:

  • The conclusion has been rearranged as suggested, and the future work has been listed in the revised manuscript.

English language: a professional proofread was employed for the language correction.

Round 2

Reviewer 3 Report

The article can be accepted